# Adverse pregnancy outcomes in women with diabetes-related microvascular disease and risks of disease progression in pregnancy: A systematic review and meta-analysis

Sophie Relph[1], Trusha Patel[2], Louisa Delaney[1], Soha Sobhy[3], Shakila Thangaratinam[4,5]*

1 Department of Women & Children's Health, King's College London, London, United Kingdom, 2 Department of Women's Health, Barnet Hospital, Royal Free NHS Foundation Trust, London, United Kingdom, 3 Barts Research Centre for Women's Health (BARC), Barts and the London School of Medicine and Dentistry, London, United Kingdom, 4 WHO Collaborating Centre for Global Women's Health, Institute of Metabolism and Systems Research, University of Birmingham, Birmingham, United Kingdom, 5 Birmingham Women's and Children's NHS Foundation Trust, Birmingham, United Kingdom

* s.thangaratinam.1@bham.ac.uk

**Data Availability Statement:** All relevant data are within the manuscript and its Supporting Information files.

## Abstract

### Background

The rise in the global prevalence of diabetes, particularly among younger people, has led to an increase in the number of pregnant women with preexisting diabetes, many of whom have diabetes-related microvascular complications. We aimed to estimate the magnitude of the risks of adverse pregnancy outcomes or disease progression in this population.

### Methods and findings

We undertook a systematic review and meta-analysis on maternal and perinatal complications in women with type 1 or 2 diabetic microvascular disease and the risk factors for worsening of microvascular disease in pregnancy using a prospective protocol (PROSPERO CRD42017076647). We searched major databases (January 1990 to July 2021) for relevant cohort studies. Study quality was assessed using the Newcastle–Ottawa Scale. We summarized the findings as odds ratios (ORs) with 95% confidence intervals (CIs) using random effects meta-analysis. We included 56 cohort studies involving 12,819 pregnant women with diabetes; including 40 from Europe and 9 from North America. Pregnant women with diabetic nephropathy were at greater risk of preeclampsia (OR 10.76, CI 6.43 to 17.99, $p < 0.001$), early (<34 weeks) (OR 6.90, 95% CI 3.38 to 14.06, $p < 0.001$) and any preterm birth (OR 4.48, CI 3.40 to 5.92, $p < 0.001$), and cesarean section (OR 3.04, CI 1.24 to 7.47, $p = 0.015$); their babies were at higher risk of perinatal death (OR 2.26, CI 1.07 to 4.75, $p = 0.032$), congenital abnormality (OR 2.71, CI 1.58 to 4.66, $p < 0.001$), small for gestational age (OR 16.89, CI 7.07 to 40.37, $p < 0.001$), and admission to neonatal unit (OR 2.59, CI 1.72 to 3.90, $p < 0.001$) compared to those without nephropathy. Diabetic retinopathy was associated with any preterm birth (OR 1.67, CI 1.27 to 2.20, $p < 0.001$) and preeclampsia

**Funding:** The author(s) received no specific funding for this work.

**Competing interests:** The authors have declared that no competing interests exist.

**Abbreviations:** CEMACH, Confidential Enquiry into Maternal and Child Health; CI, confidence interval; COVID-19, Coronavirus Disease 2019; ETDRS, Early Treatment Diabetic Retinopathy Study; MeSH, Medical Subject Headings; NICE, National Institute for Health and Care Excellence; OR, odds ratio.

(OR 2.20, CI 1.57 to 3.10, $p < 0.001$) but not other complications. The risks of onset or worsening of retinopathy were increased in women who were nulliparous (OR 1.75, 95% CI 1.28 to 2.40, $p < 0.001$), smokers (OR 2.31, 95% CI 1.25 to 4.27, $p = 0.008$), with existing proliferative disease (OR 2.12, 95% CI 1.11 to 4.04, $p = 0.022$), and longer duration of diabetes (weighted mean difference: 4.51 years, 95% CI 2.26 to 6.76, $p < 0.001$) compared to those without the risk factors. The main limitations of this analysis are the heterogeneity of definition of retinopathy and nephropathy and the inclusion of women both with type 1 and type 2 diabetes.

### Conclusions

In pregnant women with diabetes, presence of nephropathy and/or retinopathy appear to further increase the risks of maternal complications.

## Author summary

### Why was the study done?

- The rate of diabetes in young women is increasing, meaning that more have diabetes during pregnancy.

- Diabetes can cause complications in the kidneys (nephropathy), eyes (retinopathy), and nerves (nephropathy).

- When planning antenatal care that enables safe pregnancy in women with diabetes and its complications, both healthcare professionals and women need robust information on the magnitude of the possible risks affecting either the mother or baby, and the factors associated with worsening of the diabetic complications during pregnancy.

### What did the researchers do and find?

- The researchers reviewed all the research published on this topic between January 1990 and July 2021.

- Diabetic kidney disease significantly increased the risk of the woman having preeclampsia or a cesarean birth during pregnancy, the baby being born early, small or having abnormalities, the baby requiring neonatal care after birth or being stillborn, over and above the risk for diabetic women without kidney damage. Diabetic eye disease also increased the risk of early birth or preeclampsia.

- Pregnant women with diabetes were more likely to get new or worsening eye damage if it was their first baby, they smoked, they already had advanced eye damage, or they had had diabetes for a long time.

### What do these findings mean?

- Antenatal care of pregnant women with diabetic eye or kidney damage should involve a multidisciplinary team, including maternal medicine and kidney specialists.

- Women with risk factors identified in this review for worsening eye damage should be referred for closer monitoring during pregnancy, and specialist review where deterioration is noted.

- Further research be carried out to study long-term outcomes beyond pregnancy for women with diabetic eye or kidney complications.

## Introduction

The global prevalence of diabetes in adults doubled between 1980 and 2014 [1]. Many were diagnosed at a young age, particularly with type 2 diabetes, due to the obesity epidemic and sedentary lifestyle [2]. This trend has resulted in an increase in the numbers of reproductive aged women entering pregnancy with preexisting diabetes, with equal proportions diagnosed with type 1 and type 2 diabetes in some settings [3]. Pregnant women with long-standing diabetes are more likely to have microvascular complications manifesting as retinopathy or nephropathy [4]. Diabetic nephropathy is reported in 5% to 10% of pregnant women with type 1 diabetes [4] and about 2% to 3% with type 2 diabetes [5]. Diabetic retinopathy, the leading cause of blindness in reproductive aged women [6], affects 1 in 7 pregnant women with type 2 diabetes and almost half of pregnant women with type 1 diabetes [7]. Both nephropathy and retinopathy can worsen during pregnancy.

The recent confidential enquiries into maternal deaths across the United Kingdom highlighted that pregnant women with preexisting comorbidities are most at risk of death and major morbidity, stressing the need for accurate risk assessment and individualized management [8]. Current care of pregnant women with preexisting diabetes focuses on ascertaining the presence of microvascular complications, monitoring their progression and screening for pregnancy complications [9]. In order to plan pregnancy and optimize the antenatal management in women with preexisting diabetes and microvascular disease, both healthcare professionals and women need robust information on the magnitude of expected maternal and perinatal risks and the factors associated with deterioration of microvascular disease during pregnancy. However, existing studies are small with imprecise findings, and there are no meta-analyses to provide robust quantitative information.

We undertook a systematic review and meta-analysis to quantify the magnitude of association between the presence of diabetic nephropathy, retinopathy, and/or neuropathy on maternal and perinatal outcomes and the risk factors for microvascular disease progression during pregnancy.

## Methods

Our systematic review and meta-analysis was done using a prospective protocol (PROSPERO CRD42017076647) [10] according to current recommendations. We reported our findings as per the PRISMA guidelines (S1 Appendix) [11].

### Search strategy and selection criteria

We searched MEDLINE, Embase, and Cochrane databases (January 1990 to July 2021) without language restrictions for studies reporting maternal or perinatal outcomes in pregnant women with preexisting diabetes, with and without nephropathy, retinopathy, and/or neuropathy, and

on risk factors associated with disease progression. We used Medical Subject Headings (MeSH) headings, free-text and expanded synonyms of "diabet*" combined with "nephropath*," "retinopath*" or "neuropath*," and "pregnan*." The full search strategy is provided in S2 Appendix. We supplemented the results with a manual search of the reference lists.

Studies were selected for inclusion in 2 stages. First, we screened the titles and abstracts of all citations for potentially relevant papers. Second, we examined the full texts of these papers. Two independent reviewers (TP and SR) conducted the screening and the full-text evaluation against prespecified inclusion criteria. Any discrepancies were resolved after discussion with a third reviewer (ST). We included cohort studies if they reported on preexisting diabetes in pregnant women with and without microvascular complications (nephropathy, retinopathy, neuropathy, or any) and maternal outcomes such as early preterm birth (before 34 weeks' gestation), any preterm birth (before 37 weeks' gestation), preeclampsia or cesarean birth, and perinatal outcomes such as stillbirth, neonatal death, perinatal death, small or large for gestational age fetuses, congenital abnormalities, or admission to the neonatal unit (S3 Appendix). Stillbirth was defined by an intrauterine fetal death at or after 24 completed weeks. Neonatal death was defined as any death within 28 days of birth. Perinatal death included death by either definition. Small and large for gestational babies were those with birth weight less than the 10th centile for gestational age and above the 90th centile, respectively, using the centile definitions from the original studies. We accepted the study authors' definitions or classification systems used to define diabetic microvascular diseases and all other outcomes.

We also included studies if they reported on risk factors for microvascular disease progression or onset in pregnancy such as parity, disease severity, time since diagnosis of diabetes, smoking, or ethnicity. When assessing for retinopathy progression, we included study-reported deterioration in the severity defined by any of the following grading systems: White classification of diabetes in pregnancy (class C, D, or R), English classification (background, preproliferative or proliferative retinopathy), or the Early Treatment Diabetic Retinopathy Study (ETDRS) classification (mild/moderate/severe nonproliferative or proliferative retinopathy). When assessing for progression of nephropathy, we included progression from micro- to macroalbuminuria or to end-stage renal disease, and deterioration of renal function as assessed by creatinine clearance. We did not include review articles, guidelines, editorials, case studies and case series, or animal or in vitro studies.

## Study quality assessment

Two independent reviewers (SR and LD) undertook quality assessments of studies included in the meta-analysis using the Newcastle–Ottawa scale [12]. Studies with greater than 80% follow-up rates were awarded a star for outcome/exposure assessment. Studies were marked as having a low risk of bias if they scored 4 stars for selection, 2 stars for comparison, and 3 stars for exposure/outcome. Studies were marked as having a medium risk of bias if scored 2 or 3 stars for selection, 1 for comparison and 2 for outcome/exposure. Any study with a score of 1 or 0 for the selection and outcome/exposure assessments or 0 for the comparison assessment was deemed to have a high risk of bias.

## Data extraction and analysis

Data were extracted independently by 2 reviewers (SR and TP) onto a predesigned spreadsheet. Where data from the same population were duplicated, the data from the larger population were included. We calculated the individual study odds ratios (ORs) of adverse pregnancy outcomes in pregnant women with preexisting diabetes with and without nephropathy, with and without retinopathy, and with either or both retinopathy and nephropathy than without

any microvascular disease. The estimates were pooled using a random effects model. For continuously measured risk factors for disease progression, we computed the weighted mean difference using a random effects model. All confidence intervals (CIs) are presented at the 95% significance level. We assessed for heterogeneity between studies using $I^2$ tests. Publication bias and the effect of small studies were assessed on outcomes with at least 10 studies using funnel plots and Egger's tests [13]. Sensitivity analyses were conducted to see whether there was a different effect when only studying women with type 1 diabetes or by removing studies at a high risk of bias from the analysis. We assessed publication bias using Egger's test in Stata v16. All other analyses were undertaken using Stata SE (version 12) statistical software [14].

### Role of the funding source

There was no funding for this study. The corresponding author had full access to all the data in the study and had final responsibility for the decision to submit for publication.

## Results

Of the 2,985 citations identified, we selected 245 titles for detailed assessment; and 56 papers were included (12,819 pregnant women) (Fig 1).

### Characteristics of the included studies

The majority of the studies were from Europe (40 studies), followed by North America (9 studies), Asia (4 studies), the Middle East (2 studies), and South America (1 study). Eighteen studies reported on maternal outcomes for retinopathy and 20 for nephropathy, 13 on offspring

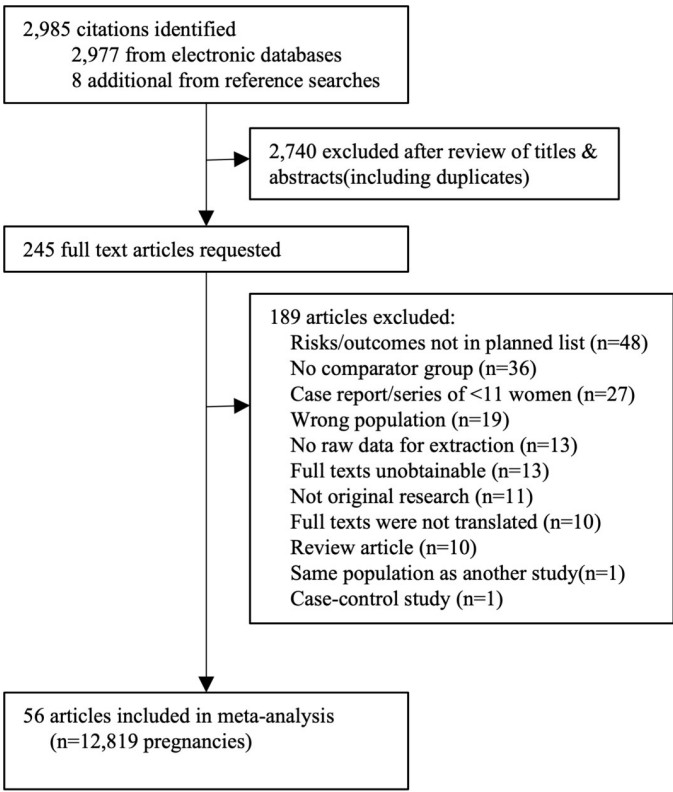

**Fig 1. PRISMA flow chart of included studies in the systematic review.**

outcomes for retinopathy and 20 for nephropathy, and 21 for risk factors for disease progression. No studies reported on outcomes for women with diabetes-related neuropathy or risk factors for progressive neuropathy during pregnancy. Twelve studies included women with both type 1 and type 2 diabetes, 30 only type 1 diabetes, 6 only insulin-dependent diabetes, 1 type 2 diabetes, and 7 unspecified diabetes type. Retinopathy was either graded using modifications of White's classification system for diabetic complications in pregnancy [15] or according to background, preproliferative, and proliferative status. Deterioration of nephropathy was described heterogeneously as either progression to nephrotic syndrome (>3 g proteinuria per day), increase in serum creatinine by >15%, creatinine clearance deterioration by >10%, elevation in creatinine greater or equal to 50% over baseline or 2-fold increase in rate of decline of glomerular filtration rate, or progression to dialysis, precluding inclusion of this outcome in meta-analysis [16–23]. The characteristics of the included studies, including inclusion and exclusion criteria, exposures, and outcomes, are provided in S4 Appendix.

## Quality of the studies

Of the 56 included studies, half (50.0%, 28/56) were considered to be at high risk of overall bias. No study had high risk of bias for adequate sample selection or for reporting outcomes. Twenty-eight studies (50.0%, 28/56) were at high risk of bias for comparability of the population. The proportion of studies deemed to have low, medium, or high risk of bias is shown in Fig 2, and details of individual study scores in S5 Appendix.

## Risk of adverse pregnancy outcomes in women with diabetic nephropathy

**Maternal outcomes.** In pregnant women with preexisting diabetes, presence of nephropathy was associated with a 10-fold increase in the risk of preeclampsia (OR 10.76, 95% CI 6.43 to 17.99, $p < 0.01$, $I^2 = 64\%$; 12 studies) [18,21,22,24–32], 6.9-fold increased risk of preterm birth before 34 weeks' (OR 6.90, 95% CI 3.38 to 14.06, $p < 0.001$, $I^2 = 47\%$; 8 studies) [5,16,18,21,22,24,27,30,33], 4.5-fold increase in preterm birth before 37 weeks' (OR 4.48, 95% CI 3.40 to 5.92, $p < 0.001$, $I^2 = 0\%$; 9 studies) [5,18,22,24,25,29,30,34,35], and

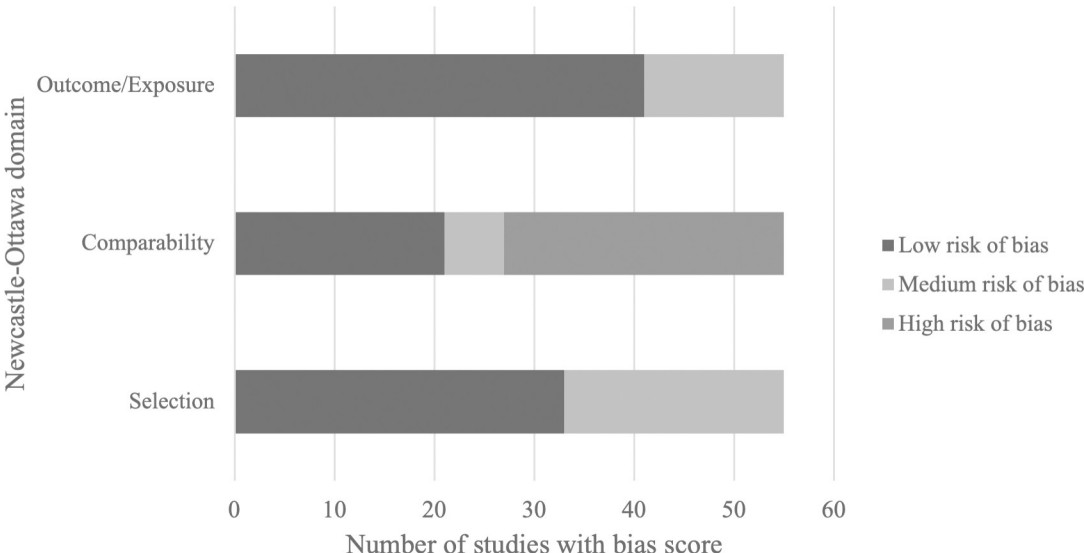

**Fig 2. Quality of the studies included in the systematic review for study selection, comparability, and ascertainment of outcome.**

increased risks of pregnancy-induced hypertension (OR 2.69, 95% CI 1.26 to 5.76, $p$ = 0.01, $I^2$ = 72%; 6 studies) [18,24,25,30–32], and cesarean section (OR 3.04, 95% CI 1.24 to 7.47, $p$ = 0.02, $I^2$ = 70%; 5 studies) [16,18,21,25,36] compared to those without nephropathy (see Fig 3). Sensitivity analyses that restricted the meta-analyses to only women with type 1 diabetes or only studies with low to medium risk of bias showed similar findings (Tables A and B in S6 Appendix).

**Perinatal outcomes.** Presence of diabetic nephropathy was significantly associated with increased risk of adverse perinatal outcomes (Fig 3) such as congenital abnormality (OR 2.71, 95% CI 1.58 to 4.66, $p$ < 0.001, $I^2$ = 0%; 6 studies) [16,18,21,22,30,37], small for gestational age fetus (OR 16.89, 95% CI 7.07 to 40.37, $p$ < 0.001, $I^2$ = 0%; 5 studies) [5,16,22,27,30], perinatal mortality (OR 2.26, 95% CI 1.07 to 4.75, $p$ = 0.03, $I^2$ = 0%; 6 studies) [18,22,25,27,30,38], and neonatal unit admission (OR 2.59, 95% CI: 1.72 to 3.90, $p$ < 0.01, $I^2$ = 10%; 2 studies) [5,25] compared to those without nephropathy; the risk of large for gestational age fetus at birth was reduced (OR 0.33, 95% CI 0.17 to 0.64, $p$ = 0.01, $I^2$ = 0%; 3 studies) [5,18,27]. One study showed no difference in the risk of low Apgar score (<7 at 1 minutes) (OR 1.68, 95% CI 0.99 to 2.87, $p$ = 0.05) or neonatal acidosis (cord arterial pH <7.05, OR 0.36, 95% CI 0.05 to 2.68, $p$ = 0.30) [25]. Sensitivity analyses (Tables A and B in S6 Appendix) restricted to only women with type 1 diabetes, or only studies with a low to moderate risk of bias, showed similar

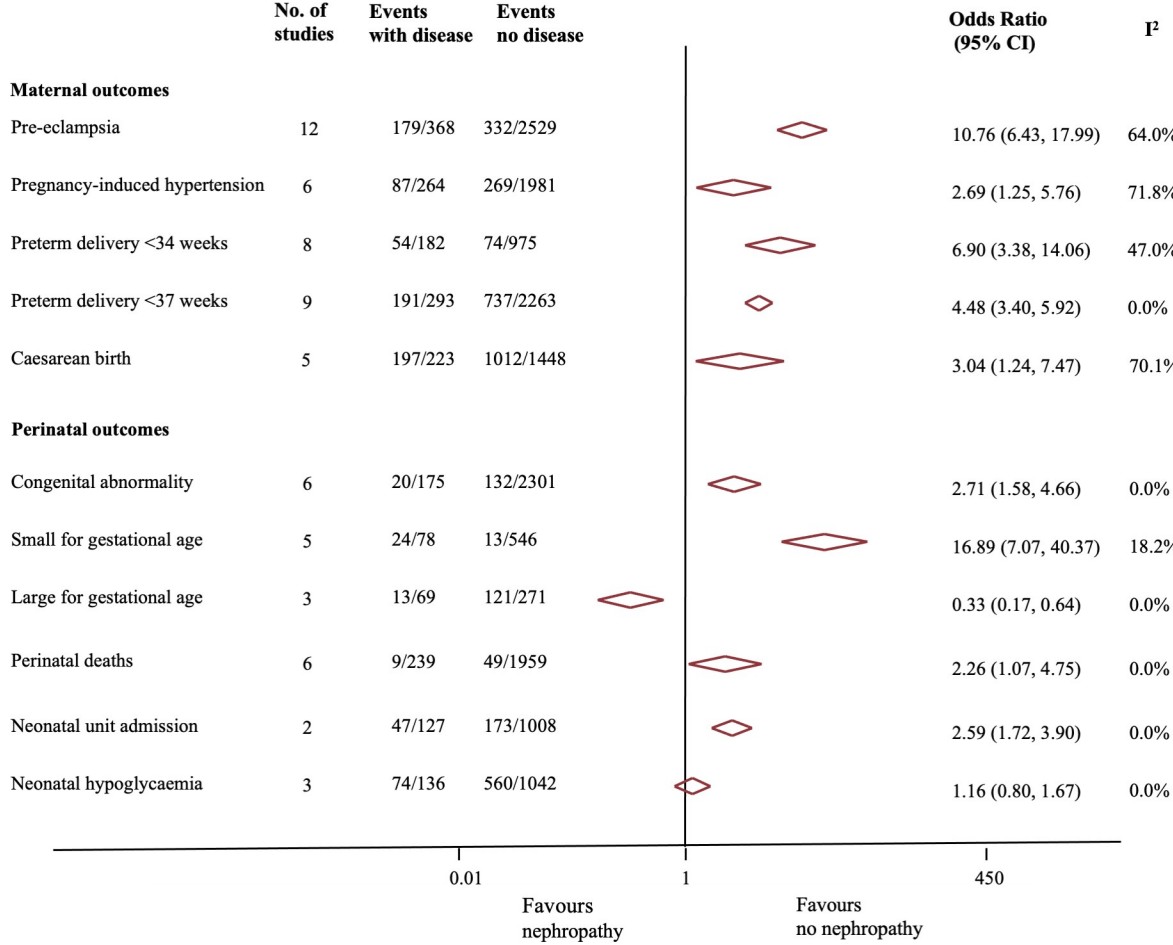

| | No. of studies | Events with disease | Events no disease | | Odds Ratio (95% CI) | $I^2$ |
|---|---|---|---|---|---|---|
| **Maternal outcomes** | | | | | | |
| Pre-eclampsia | 12 | 179/368 | 332/2529 | | 10.76 (6.43, 17.99) | 64.0% |
| Pregnancy-induced hypertension | 6 | 87/264 | 269/1981 | | 2.69 (1.25, 5.76) | 71.8% |
| Preterm delivery <34 weeks | 8 | 54/182 | 74/975 | | 6.90 (3.38, 14.06) | 47.0% |
| Preterm delivery <37 weeks | 9 | 191/293 | 737/2263 | | 4.48 (3.40, 5.92) | 0.0% |
| Caesarean birth | 5 | 197/223 | 1012/1448 | | 3.04 (1.24, 7.47) | 70.1% |
| **Perinatal outcomes** | | | | | | |
| Congenital abnormality | 6 | 20/175 | 132/2301 | | 2.71 (1.58, 4.66) | 0.0% |
| Small for gestational age | 5 | 24/78 | 13/546 | | 16.89 (7.07, 40.37) | 18.2% |
| Large for gestational age | 3 | 13/69 | 121/271 | | 0.33 (0.17, 0.64) | 0.0% |
| Perinatal deaths | 6 | 9/239 | 49/1959 | | 2.26 (1.07, 4.75) | 0.0% |
| Neonatal unit admission | 2 | 47/127 | 173/1008 | | 2.59 (1.72, 3.90) | 0.0% |
| Neonatal hypoglycaemia | 3 | 74/136 | 560/1042 | | 1.16 (0.80, 1.67) | 0.0% |

0.01  Favours nephropathy  1  Favours no nephropathy  450

**Fig 3. Association between diabetic nephropathy and adverse maternal and perinatal outcomes.**

findings, except for an even higher risk of small for gestational age fetus in the latter analysis (OR 25.75, 95% CI 10.51 to 63.08, $p < 0.01$. $I^2 = 73\%$, 4 studies) [16,22,27,30].

## Risk of adverse pregnancy outcomes in women with diabetic retinopathy

**Maternal outcomes.** Presence of retinopathy in pregnant women with preexisting diabetes was associated with increased risk (Fig 4) of preeclampsia (OR 2.20, 95% CI:1.57 to 3.10, $p < 0.001$, $I^2 = 56\%$; 8 studies) [25,26,28,31,32,39–41] and preterm birth before 37 weeks' (OR 1.67, 95% CI 1.27 to 2.20, $p < 0.01$, $I^2 = 0\%$; 4 studies) [25,35,42,43]. No significant differences were observed in the rates of cesarean birth (OR 7.37, 95% CI 0.12 to 458.28, $p = 0.34$, $I^2 = 86\%$; 2 studies) [36,44] or pregnancy-induced hypertension (OR 1.32, 95% CI 1.00 to 1.75, $p = 0.05$, $I^2 = 0\%$; 5 studies) [25,31,32,39,40], although the latter was of borderline statistical significance. When the analysis was restricted to only women with type 1 diabetes, or studies with a low to medium risk of bias, the findings remained similar (Tables A and B in S6 Appendix).

**Perinatal outcomes.** We did not observe any differences in the risk of congenital abnormality, large for gestational age fetus, perinatal death, and admission to the neonatal unit in babies born to women with versus without diabetic retinopathy (Fig 4). One study compared the pH of the umbilical arterial blood (pH <7.05, OR 1.48. 95% CI 0.50 to 4.36, $p = 0.48$ and Apgar score <7 at 1 minute of age (OR 0.72, 95% CI 0.38 to 1.37, $p = 0.31$) in women with and without retinopathy and found no difference in either outcome [25]. Findings were similar in sensitivity analyses that restricted the meta-analyses to only women with preexisting type 1 diabetes or only studies with low to medium risk of bias (Tables A and B in S6 Appendix).

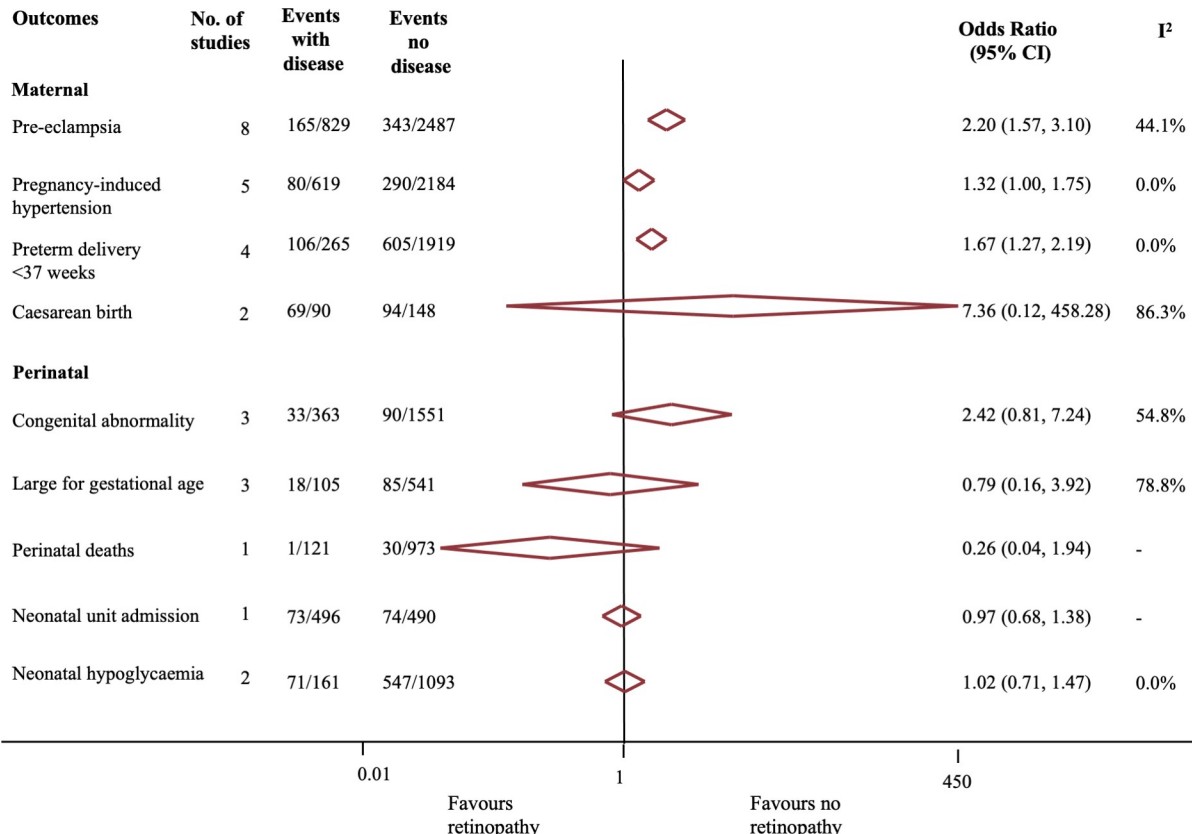

| Outcomes | No. of studies | Events with disease | Events no disease | | Odds Ratio (95% CI) | I² |
|---|---|---|---|---|---|---|
| **Maternal** | | | | | | |
| Pre-eclampsia | 8 | 165/829 | 343/2487 | | 2.20 (1.57, 3.10) | 44.1% |
| Pregnancy-induced hypertension | 5 | 80/619 | 290/2184 | | 1.32 (1.00, 1.75) | 0.0% |
| Preterm delivery <37 weeks | 4 | 106/265 | 605/1919 | | 1.67 (1.27, 2.19) | 0.0% |
| Caesarean birth | 2 | 69/90 | 94/148 | | 7.36 (0.12, 458.28) | 86.3% |
| **Perinatal** | | | | | | |
| Congenital abnormality | 3 | 33/363 | 90/1551 | | 2.42 (0.81, 7.24) | 54.8% |
| Large for gestational age | 3 | 18/105 | 85/541 | | 0.79 (0.16, 3.92) | 78.8% |
| Perinatal deaths | 1 | 1/121 | 30/973 | | 0.26 (0.04, 1.94) | - |
| Neonatal unit admission | 1 | 73/496 | 74/490 | | 0.97 (0.68, 1.38) | - |
| Neonatal hypoglycaemia | 2 | 71/161 | 547/1093 | | 1.02 (0.71, 1.47) | 0.0% |

0.01          1          450

Favours retinopathy          Favours no retinopathy

**Fig 4. Association between diabetic retinopathy and adverse maternal and perinatal outcomes.**

### Risk of adverse pregnancy outcomes in women with any diabetic microvascular complication

**Maternal outcomes.** Pregnant women with preexisting diabetes and any microvascular complication were at significantly increased risk in preeclampsia (OR 5.89, 95% CI 3.85 to 9.02, $p < 0.01$, $I^2 = 14\%$; 9 studies) [21,26,45–51], preterm birth before 34 weeks' (OR 8.49, 95% CI 1.87 to 38.63, $p = 0.01$, $I^2 = 46\%$; 3 studies) [21,47,49] and 37 weeks' (OR 2.29, 95% CI 1.85 to 2.83, $p < 0.01$, $I^2 = 0\%$; 7 studies) [25,26,43,47,49,50,52], and cesarean birth (OR 5.40, 95% CI 2.48 to 11.78, $p < 0.01$, $I^2 = 60\%$; 6 studies) [21,43,44,47,50,53] compared to those without any microvascular complication (Table 1). Findings were similar in sensitivity analyses that restricted the meta-analyses to only women with preexisting type 1 diabetes or only studies with low to medium risk of bias (Tables A and B in S6 Appendix).

**Perinatal outcomes.** Babies born to mothers with preexisting diabetes and any microvascular complication were at increased risk of having a small for gestational age baby (OR 2.49, 95% CI 1.12 to 5.57, $p = 0.03$, $I^2 = 0\%$; 5 studies) [45,47,49,50,53]. There were no differences between the groups for other perinatal outcomes (Table 1). Findings were similar with sensitivity analyses.

### Risk factors for progression of diabetic microvascular disease in pregnancy

**Retinopathy.** In pregnant women with preexisting diabetic retinopathy, the risk of disease progression or onset was significantly increased for nulliparous women (OR 1.75, 95% CI 1.28 to 2.40, $p < 0.01$, $I^2 = 0\%$; 4 studies) [18,54–56] and smokers (OR 2.31, 95% CI 1.25 to 4.27, $p = 0.01$, $I^2 = 0\%$; 5 studies) [54,57–60] (Table 2). Pregnant women with progressive diabetic retinopathy had a mean 4.51 additional years since diabetes diagnosis (weighted mean difference 4.51 y, 95% CI 2.26 to 6.76 y; $p < 0.01$, $I^2 = 78.9\%$, 7 studies) than women without progressive retinopathy [55,61–66]. There were no differences in age (WMD −0.22 years, 95% CI −0.86 to 0.42, $p = 0.50$, $I^2 = 0.0\%$, 7 studies) [55,56,60,62,63,65,66] or BMI (WMD 0.06 kg/m$^2$, 95% CI −1.05 to 1.16, $p = 0.92$, $I^2 = 1.2\%$, 4 studies) [56,60,63,66] between women with and without new or progressive retinopathy. Following exclusion of papers with high risk of bias, the risk of progressive retinopathy was no longer significant for smokers (OR 1.13, 95% CI

**Table 1. Association between diabetic nephropathy and/or retinopathy and adverse maternal and perinatal outcomes.**

| Outcomes | No. of studies | Events in women with disease | Events in women with no disease | OR (95% CI) | p-value | $I^2$ |
|---|---|---|---|---|---|---|
| Maternal outcomes | | | | | | |
| Preeclampsia | 9 | 92/273 | 95/1,196 | 5.89 (3.85, 9.02) | <0.01 | 13.8% |
| Pregnancy-induced hypertension | 2 | 26/172 | 15/455 | 2.33 (0.10, 51.95) | 0.59 | 86.7% |
| Preterm birth <34/40 | 3 | 19/73 | 17/371 | 8.49 (1.87, 38.63) | 0.01 | 64.9% |
| Preterm birth <37/40 | 7 | 253/502 | 575/1,882 | 2.29 (1.85, 2.83) | <0.01 | 0.0% |
| Caesarean birth | 6 | 210/248 | 341/769 | 5.40 (2.48, 11.78) | <0.01 | 60.4% |
| Perinatal outcomes | | | | | | |
| Congenital abnormality | 3 | 30/102 | 58/245 | 1.30 (0.75, 2.26) | 0.35 | 0.0% |
| Small for gestational age fetus | 5 | 21/349 | 11/505 | 2.49 (1.12, 5.57) | 0.03 | 0.0% |
| Large for gestational age fetus | 5 | 129/284 | 185/447 | 1.10 (0.80, 1.52) | 0.54 | 0.0% |
| Perinatal death | 5 | 18/733 | 24/1,218 | 2.85 (0.46, 17.77) | 0.26 | 75.9% |
| Neonatal unit admission | 2 | 129/643 | 149/648 | 1.21 (0.91, 1.63) | 0.19 | 0.0% |
| Neonatal hypoglycemia | 4 | 53/145 | 146/514 | 1.09 (0.56, 2.14) | 0.80 | 55.6% |

CI, confidence interval; OR, odds ratio.

**Table 2. Risk factors for worsening or new onset retinopathy in pregnant women with preexisting diabetes.**

| Risk factors | No. of studies | Events in women with risk factor | Events in women without risk factor | OR (95% CI) | p-value | I² |
|---|---|---|---|---|---|---|
| Any existing retinopathy | 15 | 283/817 | 284/1,163 | 2.64 (1.47, 4.75) | <0.01 | 79.7% |
| Background/preproliferative retinopathy | 4 | 57/163 | 35/186 | 1.94 (0.69, 5.42) | 0.21 | 60.2% |
| Proliferative retinopathy | 7 | 31/73 | 89/311 | 2.12 (1.11, 4.04) | 0.02 | 12.1% |
| Macular edema | 2 | 4/14 | 29/154 | 1.54 (0.46, 5.14) | 0.49 | 0.0% |
| Previous photocoagulation | 3 | 6/48 | 42/174 | 0.85 (0.16, 4.67) | 0.85 | 65.9% |
| Nephropathy | 4 | 32/95 | 171/754 | 1.68 (1.05, 2.69) | 0.03 | 0.0% |
| White ethnicity | 2 | 101/230 | 132/160 | 1.90 (0.76, 4.73) | 0.17 | 0.0% |
| Nulliparity | 4 | 124/404 | 166/576 | 1.75 (1.28, 2.40) | <0.01 | 0.0% |
| Smoking | 5 | 24/60 | 92/396 | 2.31 (1.25, 4.27) | <0.01 | 0.0% |

CI, confidence interval; OR, odds ratio.

0.17 to 7.59, $p = 0.90$, I² = 63%, 2 studies) [54,59]. The estimates from all other sensitivity analyses were similar (Table C in S6 Appendix).

We found a greater risk of disease progression in pregnant women with retinopathy (OR 2.64, 95% CI 1.47 to 4.75, $p < 0.01$, I² = 79.7%; 15 studies) [54–56,60–71], and nephropathy (OR 1.68, 95% CI 1.05 to 2.69, $p = 0.03$, I² = 0.0%, 4 studies) [54–56,71], at the time of the first antenatal consultation compared to those without the diagnoses. No studies reported progression to blindness. Presence of baseline background or preproliferative retinopathy (OR 1.94, 95% CI 0.69 to 5.42, $p = 0.21$, I² = 0%; 4 studies) was not found to affect retinopathy progression) [54,62,67,72]. Pregnant women with preexisting proliferative retinopathy (OR 2.12, 95% CI 1.11 to 4.04, $p = 0.02$, I² = 12.1%; 7 studies) [55,62,67,69–72], have a 2.1-fold greater risk of progression compared to women with lesser (no or background/preproliferative, respectively) changes at baseline. Proliferative retinopathy was not maintained as a risk factor after excluding studies with high risk of bias (OR 1.44, 95% CI 0.64 to 3.24, $p = 0.38$, I² = 0%, 2 studies) [55,71] or when including women with type 1 diabetes only (OR 2.04, 95% CI 0.89 to 4.67, $p = 0.09$, I² = 41.5%, 6 studies) [55,62,67,69–71]. The same was true for any retinopathy when studies at high risk of bias were excluded (OR 2.00, 95% CI 0.85 to 4.71, $p = 0.11$, I² = 82.9%, 6 studies) [54–56,64,68,71]. Following removal of studies at high risk of bias, previous photocoagulation was found to be protective of deteriorating retinopathy (OR 0.23, 95% CI 0.06 to 0.99, $p = 0.049$, 1 study) [56]. Estimates from all other sensitivity analyses were similar (Table C in S6 Appendix).

## Nephropathy

Two studies assessed the risk of antenatally deteriorating renal function among women with preexisting diabetic nephropathy compared to women without [21,22]. With no overall consensus on defining renal function deterioration, clinical heterogeneity between studies was too great to conduct meta-analysis. No studies assessed the risk of deterioration to end-stage renal failure or the risk contributed by any of the predefined maternal characteristics on antenatal progression of nephropathy.

## Publication bias

There was evidence of small study effect (Egger's test of asymmetry) for progression of retinopathy ($p = 0.003$) and preeclampsia ($p = 0.001$). The funnel plots are included in S7 Appendix.

## Discussion

### Main findings

Pregnant women with preexisting diabetes and microvascular disease (such as nephropathy and retinopathy) are at even greater risk of adverse maternal outcomes, particularly preterm birth and preeclampsia than those without microvascular complications. Mothers with diabetic nephropathy are also at high risk of offspring complications such as congenital malformations, small for gestational age fetus, and perinatal death than those without nephropathy. Nulliparity, smoking, and proliferative retinopathy at baseline are risk factors for worsening or onset of retinopathy in pregnancy.

### Comparison with existing literature

Despite increasing numbers of pregnant women with preexisting diabetes presenting with microvascular complications, no meta-analysis has been published in this area. The few available systematic reviews are narrative and mainly provided noncomparative estimates of maternal–fetal pregnancy outcomes in women with preexisting diabetic nephropathy [73]. There are no systematic reviews on pregnancy outcomes in women with diabetic retinopathy. Other narrative reviews have focused mainly on progression of retinopathy rather than on its impact on pregnancy.

The recent NICE Diabetes in Pregnancy guidelines used noncomparative data from a systematic review of 681 women to report pregnancy outcomes in women with diabetic nephropathy [9]. The Australian Diabetes in Pregnancy Society guidelines also comment on the same outcomes from a narrative review with 3 small studies [74]. The American College of Obstetricians and Gynecologists guidelines only refer to the greater risk of preeclampsia among women with diabetic nephropathy [75]. Our meta-analysis provides comparative estimates with an up to 5-fold higher sample size, and reports increased risks of additional key outcomes such as perinatal death and congenital abnormality.

Current UK guidance on screening for the small for gestational age fetus in pregnant women with diabetic microvascular complications is based on a single study [26]. Our meta-analysis includes numerous studies for this outcome and provides robust and precise estimates. The Confidential Enquiry into Maternal and Child Health (CEMACH) report in the UK found no association between presence of retinopathy and poor pregnancy outcome defined as congenital anomaly or perinatal death [76]. We found diabetic retinopathy to be associated with preeclampsia and preterm birth, which are major risk factors for maternal and perinatal morbidity. Unlike the CEMACH report with 442 women, our evidence base is larger with more robust estimates. We found an increase in the rates of congenital abnormalities in women with diabetic nephropathy. It is possible that the poor glycemic control observed in women with nephropathy is more important than the disease itself in contributing to the adverse outcome.

Both the UK National Institute for Health and Care Excellence (NICE) and Australian diabetes guidelines identified severe retinopathy at conception, duration of diagnosed diabetes, poor glycemic control and hypertension as risk factors for antenatal progression of retinopathy, and disease severity at conception for worsening of nephropathy [9,74]. Many of these conclusions were based on single studies. In addition to severity of retinopathy and duration of diabetes, our meta-analysis found additional risk factors such as nulliparity and smoking for worsening of the disease. While we did not include hypertension and glycemic control in our prespecified list of risk factors (S3 Appendix), our search strategy did capture such studies. There were 6 studies identified, which assessed how glycemia control affected progression of retinopathy [62,72,77–80], one of which also assessed progression of nephropathy [79].

Glycemic control was defined heterogeneously (hypoglycemia, mean HbA1C, or change in HbA1C measured at different time points—preconception, first, second, or third trimester) and therefore would not have been amenable to meta-analysis. Five of the same studies also studied blood pressure as a risk factor for progressive microvascular disease [62,72,77–80], with the same limitation of heterogeneity (risk factors were mean diastolic blood pressure, mean systolic blood pressure–time points varied, use of antihypertensive medication, chronic hypertension, gestational hypertension).

## Strengths and limitations

To our knowledge, ours is the largest and most comprehensive meta-analysis to date that quantifies the risk of adverse maternal and perinatal outcomes in pregnant women with diabetic microvascular complications and the risk factors for disease progression. We undertook the review with a prospective protocol in line with current recommendations, identified the studies with a detailed search strategy and without any language restrictions, evaluated the quality of the included studies, and performed appropriate meta-analyses with assessment of statistical heterogeneity. We reported the strength of association between diabetic microvascular complications and pregnancy outcomes separately and in combination and studied all clinically relevant outcomes. The findings were homogeneous for the risk of perinatal outcomes in women with diabetic nephropathy and for most maternal outcomes with retinopathy. Our sensitivity analyses allowed us to assess the robustness of our findings by excluding low-quality studies and limiting to studies that reported risks for women with type 1 diabetes separately.

There are limitations in our systematic review. Studies varied in their definitions of diabetic nephropathy and retinopathy and included women with both type 1 and type 2 diabetes. But our sensitivity analysis including only women with type 1 diabetes showed findings similar to the overall estimates. While many studies reported on the risks of diabetic retinopathy or nephropathy, none assessed neuropathy in detail. We were also unable to take into account other factors such as maternal age, BMI, and previous obstetric history that may have influenced the association between diabetic microvascular complications and pregnancy outcomes. Unlike retinopathy with comparable classification systems, the severity of nephropathy was reported variedly, which refrained us from identifying the risk factors for worsening disease. We only included studies published since 1990, but it is possible that some of the outcomes could be influenced by the variations in clinical practice over time and between institutions. There were very few events for some of the reported outcomes studied. This is reflected in the imprecision of the point estimates. Furthermore, the variations in the definitions of the populations, retinopathy, and nephropathy may have contributed to the high heterogeneity observed for some findings. We were only able to assess publication bias for 2 outcomes, because all other outcomes were assessed in less than 10 studies. Of the 2 outcomes where this was possible, we found evidence of small sample bias, suggesting that the results should be interpreted with caution. We limited our analysis of disease progression to only pregnancy as observational studies suggest that diabetic retinopathy deteriorates more rapidly in pregnancy [9,67,77], but the findings may not necessarily translate into worse long-term retinopathic severity when compared to women who were not pregnant [81].

## Implications for clinical practice

The maternal and offspring risks are significantly increased for pregnant women with diabetic nephropathy, in particular with over 10-fold increase in the risks of preeclampsia and small for gestational age fetuses. It is essential that antenatal care of pregnant women with microvascular disease should involve a multidisciplinary team, including maternal medicine and nephrology

specialists. It is possible that a higher dose (150 mg) antenatal aspirin, instead of 75 mg, may mitigate the risks of preeclampsia in women with diabetes-related microvascular disease [82]. During the Coronavirus Disease 2019 (COVID-19) pandemic, pregnant women with preexisting diabetes are also in the highest risk groups for becoming severely unwell from COVID-19, and, furthermore, there have been widespread restrictions on maternity services [83]. It is essential that the women at highest risk of adverse pregnancy outcomes are identified so that scarce resources can be appropriately targeted.

In the UK CEMACH report, pregnant women with preexisting diabetes and poor pregnancy outcomes were also less likely to have retinal assessment at the time of the first antenatal consultation than those with good outcomes [76], indicative of suboptimal diabetes care linked to suboptimal retinal monitoring. Many factors contributing to progression of retinopathy during pregnancy are modifiable such as preconception control of glucose and smoking cessation. Women with risk factors identified in this review for deteriorating eye disease should be referred for close monitoring during pregnancy [9], and specialist review where deterioration is noted.

### Recommendations for research

Further research is needed to better understand the associations between diabetic microvascular diseases and pregnancy outcomes and whether this differs by type 1 or type 2 diabetes. Although the risks of preterm birth are increased in women with diabetic nephropathy and in those with retinopathy, it is unclear whether these were spontaneous or iatrogenic preterm births; this needs to be delineated. Depending on the cause for preterm birth, further research is needed to identify effective interventions to prevent spontaneous preterm birth or to reduce the risk of iatrogenic prematurity. We need consensus on criteria for deterioration of diabetic nephropathy in pregnancy, or preferred method for the assessment of renal function decline, to identify the proportion of women with worsening disease and the risk factors for disease progression. There also needs to be a standardized method of assessing both glycemic and blood pressure control throughout pregnancy, to determine what effect these have on progression of microvascular complications. Techniques worthy of further assessment for this purpose include continuous measurement of blood pressure over 12- or 24-hour periods or provision of validated monitors for home monitoring. The paucity of evidence on obstetric and disease-related outcomes for diabetic neuropathy, including autonomic neuropathies (e.g., gastroparesis) can be addressed by systematically collecting this information in the national surveillance or registry systems such as the UK Obstetric Surveillance System or Diabetes in Pregnancy audit [84]. We need data on long-term outcomes beyond pregnancy for women with diabetic microvascular complications and their babies to obtain a comprehensive overview of the risks. This information is critical to predict the risk of disease progression during pregnancy and postnatally.

### Conclusions

Pregnant women with preexisting diabetes and microvascular diseases such as nephropathy and retinopathy are at greater risk of preeclampsia and preterm birth than those without the microvascular diseases. Women with diabetic nephropathy are also at higher risk of most major maternal and perinatal complications including congenital abnormalities, growth-restricted fetuses, and perinatal death. Nulliparous mothers, smokers, and women with retinopathy at baseline have an increased risk of deteriorating diabetic eye disease in pregnancy. Pregnant women with diabetic microvascular complications require management in speciality multidisciplinary teams with frequent, targeted antenatal care surveillance and interventions to improve maternal and perinatal outcomes.

## Supporting information

**S1 Appendix. PRISMA checklist.**
(DOCX)

**S2 Appendix. Search strategy.**
(DOCX)

**S3 Appendix. List of a priori outcomes and risk factors.**
(DOCX)

**S4 Appendix. Table of included studies.**
(DOCX)

**S5 Appendix. Scores given to included studies using the Newcastle–Ottawa scale for assessment of potential bias.**
(DOCX)

**S6 Appendix. Results of sensitivity analyses.** Table A: Maternal and perinatal outcomes in women with type 1 diabetes and vasculopathy—A sensitivity analysis. Table B: Maternal and perinatal outcomes in women with diabetes and vasculopathy—A sensitivity analysis excluding papers with high risk of bias. Table C: Risk factors for disease progression (retinopathy)—Sensitivity analyses.
(DOCX)

**S7 Appendix. Funnel plots for publication bias assessments.**
(DOCX)

**S8 Appendix. Data supporting meta-analysis on disease outcomes.**
(XLSX)

**S9 Appendix. Data supporting meta-analysis on maternal and fetal outcomes.**
(XLSX)

## Acknowledgments

The authors wish to thank Dr. John Allotey for assisting them with the evaluation for publication bias.

## Author Contributions

**Conceptualization:** Shakila Thangaratinam.

**Data curation:** Sophie Relph, Trusha Patel.

**Formal analysis:** Sophie Relph, Trusha Patel, Louisa Delaney, Soha Sobhy.

**Investigation:** Sophie Relph, Trusha Patel.

**Methodology:** Sophie Relph, Trusha Patel, Soha Sobhy, Shakila Thangaratinam.

**Project administration:** Sophie Relph.

**Resources:** Shakila Thangaratinam.

**Software:** Soha Sobhy.

**Supervision:** Soha Sobhy, Shakila Thangaratinam.

**Validation:** Sophie Relph.

**Writing – original draft:** Sophie Relph.

**Writing – review & editing:** Sophie Relph, Trusha Patel, Louisa Delaney, Soha Sobhy, Shakila Thangaratinam.

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
