## [Editor Report · Decision Letter 0]

27 May 2021

Dear Dr Relph, 

Thank you for submitting your manuscript entitled "Adverse pregnancy outcomes in women with diabetes-related microvascular disease and risks of disease progression in pregnancy: a systematic review and meta-analysis" for consideration by PLOS Medicine.

Your manuscript has now been evaluated by the PLOS Medicine editorial staff and I am writing to let you know that we would like to send your submission out for external peer review.

Please re-submit your manuscript within two working days, i.e. by May 31 2021 11:59PM.

Kind regards,

Beryne Odeny

Associate Editor

PLOS Medicine

---

## [Decision Letter · Decision Letter 1]

2 Aug 2021

Dear Dr. Relph,

Thank you very much for submitting your manuscript "Adverse pregnancy outcomes in women with diabetes-related microvascular disease and risks of disease progression in pregnancy: a systematic review and meta-analysis" (PMEDICINE-D-21-02305R1) for consideration at PLOS Medicine. 

[LINK]

In light of these reviews, I am afraid that we will not be able to accept the manuscript for publication in the journal in its current form, but we would like to consider a revised version that addresses the reviewers' and editors' comments. Obviously we cannot make any decision about publication until we have seen the revised manuscript and your response, and we plan to seek re-review by one or more of the reviewers. 

We expect to receive your revised manuscript by Aug 23 2021 11:59PM. Please email us (plosmedicine@plos.org) if you have any questions or concerns.

We look forward to receiving your revised manuscript. 

Sincerely,

Beryne Odeny, 

PLOS Medicine 

plosmedicine.org

1) Abstract:

a) Please report your abstract according to PRISMA for abstracts, following the PLOS Medicine abstract structure, available at http://www.plosmedicine.org/article/info:doi/10.1371/journal.pmed.1001419

i) Please structure your abstract using the PLOS Medicine headings (Background, Methods and Findings, Conclusions).

ii) Please combine the Methods and Findings sections into one section, “Methods and findings”. 

iii) Please revise the subheading “Interpretation” to “Conclusions.”

b) Please ensure that all numbers presented in the abstract are present and identical to numbers presented in the main manuscript text.

c) Please include p values in addition to 95% CIs

d) In the last sentence of the Abstract Methods and Findings section, please describe the main limitation(s) of the study's methodology.

e) Please address the study implications, stating what is new, without overreaching what can be concluded from the data; the phrase "In this study, we observed ..." may be useful.

2) Author summary - At this stage, we ask that you reformat your non-technical Author Summary. The Author Summary should immediately follow the Abstract in your revised manuscript. This text is subject to editorial change and should be distinct from the scientific abstract. The summary should be accessible to a wide audience that includes both scientists and non-scientists. Please see our author guidelines for more information: https://journals.plos.org/plosmedicine/s/revising-your-manuscript#loc-author-summary.

3) Please update your search to the present time.

4) Please evaluate evidence of publication bias. 

5) PRISMA checklist: 

a) when completing the checklist, please use section and paragraph numbers, rather than page numbers.

6) In the Methods and Results section:

a) Please provide 95% CIs and p values for estimates in the main text and tables

b) Please name Fig 1, “PRISMA flow chart…”

7) In discussion section, please move the “strengths and limitations” paragraph further down and place before “Implications for clinical practice”. 

8) Please use bars and whiskers in the forest plots (Figure 3 & 4), and indicate in the figure caption the meaning of the bars and whiskers

9) Please replace "Caucasian" with "white" throughout the paper.

10) Please ensure standard reference formatting:

a) Please use the "Vancouver" style for reference formatting, and see our website for other reference guidelines https://journals.plos.org/plosmedicine/s/submission-guidelines#loc-references. 

b) Please use the PLOS Medicine style reference call outs throughout the text, noting the absence of spaces within the square brackets, e.g., "... between 1980 and 2014 [1,2]."

c) Please ensure that weblinks are current and include date of access.

Comments from the reviewers:

Reviewer #1: I confine my remarks to statistical aspects of this paper. The general approach is fine but I do have a couple issues to resolve before i can recommend publication

General: When I^2 is high, it's good to try to figure out why.

p. 7 line 128-132 Why these cutoffs? This means that, even if the sample is typical, 20% will be called either small or large. It would be better if there were substantive reasons and values for weight that mark abnormality or difficulty.

Fig 2 "Number of studies" - this seems like it is a % of studies, not an N. 

 A parallel box plot seems better suited for these data

 Don't divide quality into 3 categories, use the actual quality score

Fig 3 and 4

 .01 on one side corresponds to 100 on the other, so those two should be marked

 it would be good to mark more than two points, e.g. have .01, .05, .1, ,5, 1, 2, 5, 10, 20, 100 or something like thatr

 Maybe put the ORs and p values in a table, rather than here, to allow more room for the graph

 Consider using landscape view for the same reason

Peter Flom

Reviewer #2: As the authors say, this appears to be the only major quantitative as opposed to narrative Syst. Rvw available of this growing issue. It is therefore important work. The results are dramatic if with wide confidence intervals (CI) reflecting the relatively small nos of studies & also events. However, the global importance both clinically & to Public Health are clear; obesity -T2 Diabetes (still rather under-represented, as here, it seems)- complications earlier in life and now pregnancy continue as a major threat.

I have no major methodological points but some minor ones:

- Abstract: points worth considering/ amending

i. They review papers from 1990-2020; were there hints or overt signs of time trends as identifying these complications pre- or in pregnancy improved?

ii. Numbers of events in these high Odds Ratios, as with the nos. of studies with each complication (nephropathy/ retinopathy etc), were relatively SMALL, obviously contributing to the wide CIs as above - they can say so. These CI should be only 1 decimal point throughout the text - false precision beyond.

iii. Despite point ii, under 'Interpretation' here, they are entitled to say '..further greatly increase..risks..';

iv. There is no mention here in the Abstract of why they consider 'diabetes' as a whole = that is the no of studies including Type 2DM were still few.

Results: Again include here the N for Type 1 vs/+ Type 2 studies;

Discussion: 2nd line (no.326) - they could add 'even' as '…at EVEN greater risk..', given that the additive risk of pre-eclampsia is already much higher than without DM. They highlight the current confusion over how blood pressure features due to highly variable classification etc.; they might make rationalising this a central recommendation - AND that other blood vessel measures are available for analysing the excess risk (aortic PWV/ carotid / basic cardiac measures - all surely ready for clinical testing as risk indicators (see doi.org/10.1002/uog.19021).

Refs: Their ref. 83 presumably is 'in press'.

Figure 3: the lower risk for nephropathic mothers of LGA infants (OR 0.33 there) contradicts the similar line in Table 2 (between lines 279-280) with an OR of 1.1, admittedly including retinopathic mothers also, with larger Ns. Is this discrepancy correct? If so, it emphasises the instability of the ORs generally - which they might point out themselves (? In the Abstract & Discussion/ Limitations).

Reviewer #3: PMEDICINE-D-21-02305R1. Drs. Sophie Relph et al. 2021. "Adverse pregnancy outcomes in women with diabetes-related microvascular disease and risks of disease progression in pregnancy: A systematic review and meta-analysis."

Summary of the research and reviewer's overall impression.

Summary: Focusing on pregnancy outcomes in women with preexisting diabetes and microvascular disease, the authors made a critical, prospective, concise, and incisive review of the literature and performed a meta-analysis of banked data from 12,320 pregnancies in 55 studies spanning almost 30 years. The great majority of those studies (~87%) were conducted in Europe and North America. Two independent reviewers screened papers for inclusion, and a third independent reviewer adjudicated conflicts. The 55 reports were winnowed from 2,811 citations initially selected. Results are expressed as odds ratios (OR).

Considering pregnancies with both preexisting maternal nephropathy and diabetes, compared to those with diabetes only:

* Mothers were at greatly increased OR risk of both early (<34 weeks) and any preterm delivery.

* Offspring were at increased OR risk of congenital abnormalities, being small for gestational age (SGA), having needed admission to the NICU, and perinatal death.

Similarly, preexisting diabetic retinopathy in moms was associated with increased OR risk of preeclampsia and preterm delivery. 

Morbidity of retinopathy was elevated in moms with nulliparity*, smoking, preexisting proliferative disease, and prior duration of diabetes.

Doctor Relph and her colleagues conclude that "In pregnant women with diabetes, presence of nephropathy and/or retinopathy further increase[s] the risks of maternal complications."

Reviewer's summary statement: This manuscript provides a helpful and timely primer on a growing and difficult health challenge, worldwide. It is an enjoyable read. A complex primary literature is artfully distilled into understandable queries into the problem at hand. Quality measures are explicit (lines 146-154; Figure 2). This new meta-analysis, the largest such study to date, provides clear, evidence-based guidance that obstetric and endocrine clinicians need to intervene early and aggressively in these at-risk women, for the sake of both the moms and their children. The authors' synthesis succeeds in being quantitative, rather than merely narrative. This reviewer predicts that the present work will be highly cited, and that it will help to inform favorable changes in medical policy and practice.

Disclosure regarding this reviewer's qualifications: This basic-science investigator studies metabolism of the major human metabolic diseases, including gestational diabetes. This reviewer is neither a clinician nor a statistician.

Evidence and examples. Specific questions and concerns:

This manuscript rightly focuses on short-term outcomes in mother and child. In the few longer-term studies that have been published, what are the later-in-life consequences of women becoming pregnant while they are both diabetic and afflicted with microvascular disease?

Definitions of most patient characteristics and medical conditions are explicitly given in the text (e.g., lines 120-143, 181-198). Multicenter, observational data considered here span more than 30 years of medical practice, and the clinical definitions of several concepts evolved over the decades concerned. *For example, use of the adjective "nulliparous" and related terms here (abstract, last sentence of Findings; Table 2; Leperco entry in page 9 of the appendices; and in manuscript lines 62, 290, 330, 399, and 457) might confuse some readers because its definition has varied across time and space—is it meant to signify "mothers who for the first time carried pregnancies to term" here?

Institutional review boards and informed consent by human subjects: Pursuant to the PLoS Medicine's guidelines (pages 4 and 6, left sidebars), could the authors please provide an appendicular table that summarizes the names of the Institutional Review Boards with authority over the 55 observational studies included here, and noting whether or not the original, on-site investigators confirmed that the subjects' consent was obtained, consistent with the Declaration of Helsinki, or some similar international standard?

Heavy weighting toward Euro-North American studies (lines 182-183) should be noted briefly in the abstract. Outcomes could well be different in, say, South-Asian and Chinese populations.

Association scores between diabetic retinopathy and Caesarean birth are strikingly broad (Figure 4, fourth line down). Could the authors please comment on this briefly in the text? Does this relate to how medical practices differ among the geographic regions under consideration?

Authors acknowledge (e.g., lines 348-351) that limitations of many of the 55 constitutive publications do not allow a thorough stratification of pregnancy outcomes in Type 1 versus Type 2 diabetes, which would have been helpful, and should be addressed in future studies.

Other point / minor copy edit.

Meaning of the phrase "at booking" (lines 304 and 427) might not be readily apparent to some readers whose first language is not British English.

[LINK]

---

## [Decision Letter · Decision Letter 2]

7 Oct 2021

Dear Dr. Relph,

Thank you very much for re-submitting your manuscript "Adverse pregnancy outcomes in women with diabetes-related microvascular disease and risks of disease progression in pregnancy: a systematic review and meta-analysis" (PMEDICINE-D-21-02305R2) for review by PLOS Medicine.

I have discussed the paper with my colleagues and the academic editor and it was also seen again by three reviewers. I am pleased to say that provided the remaining editorial and production issues are dealt with we are planning to accept the paper for publication in the journal.

[LINK]

We look forward to receiving the revised manuscript by Oct 14 2021 11:59PM.   

Sincerely,

Beryne Odeny, 

PLOS Medicine

plosmedicine.org

Requests from Editors:

Thank you for responding to our initial comments. Before we proceed, please address the following:

1) Please remove “Funding” from the abstract. 

 - Please remove the “Data availability statement” and “Declaration of interest” from the end of the main text. In the event of publication, this information will be published as metadata based on your responses to the submission form.

2) Please do not report P<0.01; report as P < 0.001.

3) Please place in-text reference call outs prior to punctuation noting the space between the last word and call out. For example, “…countries [1,2].”

a) Please ensure that journal name abbreviations consistently match those found in the National Center for Biotechnology Information (NCBI) databases. https://journals.plos.org/plosmedicine/s/submission-guidelines#loc-references. 

b) Please include access date for ref #10, 12 & 76.

4) For additional emphasis, please pay special attention to reviewer #2’s comments.

Comments from Reviewers:

Reviewer #1: The authors have addressed my concerns and I now recommend publication.

Peter Flom

Reviewer #2: The combination of T1 and Y2 diabetes in this Review is too important to omit from the Abstract. The authors state the omission is due to space limits. However, making room is simple. For example, if they insert 'Type 1 or 2' in line 47 of their Abstract's text in front of 'diabetic microvascular disease' then in line 53, remove 'of these were' in front of 'Europe', and remove the 'and' after 'Europe', replacing it with a comma, the word count remains the same as now.

This reviewer does not accept that confidence intervals should be to 2 decimal points when throughout, the number of events are frequently small, and types of diabetes are combined. As previously in the reviewer's point, now no.23, with respect it is false precision and not accuracy to quote such detail. The editors can advise.

Finally, and surely, the job of a systematic review when risks such as these are so high is not just to say so but to indicate where further work could usefully perhaps reduce them. While blood pressure (BP) may be important, it is obviously imprecise especially when taken in a clinic. More modern techniques, which need not be complex and appear to be more stable, are available, as cited previously. Testing these out could offer more than badly taken, routine BP, as could home BP measures, or occasionally 12 or 24 hour values. Similarly simple retinal or renal tests, advised and perhaps tested by collaborating specialists they suggest to involve, might be tried. Commenting that they are not worth mentioning because they are not currently routine would mean nothing novel would ever get introduced.

Reviewer #3: Doctors Relph and colleagues made thoughtful responses to the editor's and reviewers' comments.

Reassuringly, their updated meta-analysis supported their earlier conclusions.

In-depth queries regarding statistics and the authors' responses thereto are largely beyond this reviewer's competence and experience.

Regarding point 35, this reviewer understands, but respectfully disagrees that an on-line compendium of IRB and similar ethical - review - board approvals is not essential here.

[LINK]

---

## [Editor Report · Decision Letter 3]

26 Oct 2021

Dear Dr Relph, 

On behalf of my colleagues and the Academic Editor, Dr. Jenny E Myers, I am pleased to inform you that we have agreed to publish your manuscript "Adverse pregnancy outcomes in women with diabetes-related microvascular disease and risks of disease progression in pregnancy: a systematic review and meta-analysis" (PMEDICINE-D-21-02305R3) in PLOS Medicine.

PRESS

Sincerely, 

Beryne Odeny 

PLOS Medicine